# Dreams Shared on Social Networks during the COVID-19 Pandemic: A Tower of Babel or Noah’s Ark?—A Group-Analytic Perspective

**DOI:** 10.3390/ijerph20043534

**Published:** 2023-02-17

**Authors:** Shulamit Geller, Gal Van den Brink, Yehoshua Akerman, Sigal Levy, Tuli Shazar, Gil Goldzweig

**Affiliations:** 1School of Behavioral Sciences, The Academic College of Tel Aviv-Yaffo, Tel-Aviv 68182, Israel; 2Statistical Education Unit, The Academic College of Tel Aviv-Yaffo, Tel-Aviv 68182, Israel

**Keywords:** dreaming, COVID-19, lockdown, SNS, group analysis, dreamtelling, trauma

## Abstract

Dream sharing is a universal practice, and various incentives have been identified, including emotional processing, emotional relief, and demands for containment. Shared dreams can contribute to an individual’s understanding of social reality during traumatic and stressful events. The present study examined dreams shared on social network sites (SNS) during the first COVID-19 lockdown, applying a group-analytic approach. A qualitative dream content analysis conducted by a group of researchers analyzed 30 dreams shared on SNS, focusing on their contents, dominant emotions, and unique group processes. The dream content analysis yielded three meaningful and coherent themes: (1) dominant threats: enemy, danger, and COVID-19; (2) emotional fusion: confusion and despair alongside recovery and hope; and (3) group processes characterized by movement between being alone and being together. The results deepen our understanding of both unique social and psychological group processes and of people’s main experiences and key psychological coping mechanisms in times of collective trauma and natural disasters. They also demonstrate the transformative potential of dreamtelling for individuals’ coping experiences and building hope through the creative social relationships formed within SNS groups.

## 1. Introduction

The COVID-19 pandemic has been shown to be associated with a significant increase in individual and community psychological distress globally, nationally, and internationally, e.g., [1]. The risk of dying from an unfamiliar illness, affecting relatives, and financial cost has been minimized by mass infection control measures, such as isolation and social distancing. Although recent studies have reported clear negative psychological effects, including confusion, anger, and symptoms of post-traumatic stress [2], they have specifically observed, as in other episodes of collective distress [3], a strong impact on dreams, such as increased dream frequency, increased negative emotions, and bizarreness [1,4]. The traumatic occurrences led to a wealth of websites collecting reports and dream content during the first lockdown period (March to May 2020) which received extensive global media exposure.

Psychoanalytic theories originating with Freud [5] have shared the assumption that dreams play a role as mental mechanisms that process intensive emotional experiences. This concept was further expanded by theorists such as Bion [6], who theorized the dream as a “vessel”, thus allowing for the processing of matter that was not accessible before the dream. Contemporary dream theories and findings have claimed that during times of prolonged stress, dreams can assist people deal with negative emotions and distress [7,8,9]. Simultaneously, they may enable extra adaptive neural functions for emotional processing, where neural connections form more hastily than in the waking state [10]. Therefore, these connections are not created arbitrarily, nor do they mirror actual traumatic experiences but, rather, are steered by the dreamer’s emotions. Hartmann [11] asserts that this association of stressful experiences and emotions with existing memories enables “the storm to calm”. The results of research studying COVID-19-lockdown dreams in Italy indicate that focusing on the function of dreams to facilitate affective integration and elaboration can promote emotional coping [12]. Additionally, recent findings from a living systematic review aimed at providing a broad overview of dreaming during the pandemic in the general population show: (1) the intense oneiric activity associated with greater psychological distress, especially during lockdowns; (2) the unique pandemic dreams contents (e.g., masks, isolation, failures to socially distance, and fear of contagion); and (3) the specific function of dreaming as emotional regulator [13]. 

### 1.1. The Group-Analytic Approach to Dream Sharing 

Dream shared around the tribal campfire are understood from ancient times as means of bonding, learning, and coping [14,15]. Interestingly, the social network sites (SNS) through which individuals shared their vivid and intensive dreams during the pandemic [4] and which facilitated informational and emotional support from individuals and communities around the world during quarantine and social isolation have been said to replace the tribal campfire [15,16]. 

From a group-analytic viewpoint, SNS represent a web of relationships linking individuals to each other in a matrix structure [17]. The matrix—a vital concept in group analysis—is the common ground of conscious and unconscious, past, and present communications, related to the sociocultural organization and the context [18]. Two aspects of the matrix are the “foundation matrix” and the “dynamic matrix”. The foundation matrix relates to the organization of the wider society within which the group operates and contains aspects such as language, social status, historical accounts, and rituals [19]. The dynamic matrix which develops from the interface of individuals with the foundation matrix [20] is the organization of the group itself and is built on the interconnections and patterns of communication within the group. The dynamic and foundation matrices are interdependent, with all communications contributing meanings to the group process [21]. The impact of collective traumatic events is expressed and can be detected in both the foundation matrix of the contextual society and the dynamic matrix of the social system in question [22]. The dreamer and their audience can therefore be seen as encountering in a dynamic matrix (the SNS dream groups) in which dreams that are both individual creations and group property are explored by all members of the group [23]. Consistent with this conceptualization, dreams respond to collective concerns and interests within the matrix [24], but the dreamer, as the person who initiated its elaboration, has a delegated role [25]. As dreams are shared within the group, other participants become engaged in this process of containment and elaboration. 

Dreaming and dreamtelling are seen in contemporary group theories, as a form of communication, where group members request containment [26], needed for promoting novel means of thinking and understanding denied individual, social, and cultural experiences [27,28,29]. Sharing and discussing dreams in a group setting may therefore increase members’ empathy [30] and capacity to cope with baffling hectic and traumatic collective encounters as a form of social and “moral witnessing” [31]. In particular, Bermundez [32] advocated the significance of such witnessing in both containing and attaining a reflective ability for unformulated and non-symbolized collective traumatic experiences. Recent studies on pandemic dreams affirm these results, highlighting the use of greater symbolization process during dreams [1], and demonstrating that while dreams indicate mental pain, the method of examining and telling them was positively appraised by participants [33]. These conclusions denote that sharing and discussing dreams during social isolation can have a constructive impact on individuals’ experiences, assisting them to relieve mental pain and possibly guiding them to take part in other methods of creative communal relation. 

When employing the dreamtelling approach to the first COVID-19 lockdown on SNS, four phases can be characterized [26]. In the first phase, the surprise of COVID-19—a unique traumatic event and a threat on both the individual and the society—affects groups on both national and international levels. This collective preoccupation is then dreamed by the dreamer in a second phase, after which, in a third phase, the dreamer recalls the dream and self-reflects on it. In the fourth phase, when the digestion is not satisfactory (i.e., the dreamer fails to contain and elaborate the emotional tension first tackled while dreaming), the dream is written down and mailed to SNS where collaborative processing of the dream description and its resonance is made plausible. This digestion, thus, goes beyond the individual to be processed among partners, so that the dream is then “re-dreamed” by the group. Audience’s responses to the dream involve the use of exchange, mirroring, and resonance mechanisms [23] that allows them to re-dream the dream among partners and subsequently digest difficult traumatic contents. Exchange refers to the process of relieving isolation by sharing personal experiences with the group and creating an opportunity for both the individual participant and the group to express differences [19]. Mirroring is a process that occurs in a group when individuals see themselves, or part of themselves (often a repressed part), reflected in the interactions of other group members [23]. Resonance, the third mechanism, refers to spontaneous and unconscious verbal or non-verbal interaction between members in the group that vibrate and reinforce each other so that each member picks out what is relevant to them from the common pool [20]. Two main aspects of a member’s subjective experience are involved in this process: (1) extreme emotional involvement in a certain theme in the group; and (2) the feeling of being transparent to others [34]. Taken together, these three group mechanisms provide each individual member with both what is known and what is unknown or unknowable, which is then fed back into the common matrix [20].

### 1.2. The Present Study 

The present study is a novel attempt to consider both the adaptive function of dreaming [35,36], especially in times of continuing stress, and the supposition that sharing dreams in groups has the potential to contribute to both our coping and knowledge of the social reality of individuals [20,29]. It is, to the best of our knowledge, the first study to analyze dreams posted and shared on SNS during the first COVID-19 lockdown within a group context. We took an epistemologically group-analytic approach, according to which the origins of psychic difficulties are not restricted to the individual psyche but also belong to a network of interactions, namely the group matrix [18,37]. The group-analytic approach was used to explore the contents and emotions of dreams, identify the unique group processes, and portray the collective digestive function that dreamtelling on SNS serves in coping with the traumatic reactions to the first COVID-19 lockdown [24].

## 2. Materials and Methods

### 2.1. Sampling and Data Collection 

Five dream sharing groups were located on Facebook: “Covid19 Dreams”, “Dreams in Solitary”, “IASD: International Association for the Study of Dreams”, “Pandemic Dream Sharing Group”, and “350 Dreamers”. Of these groups, two were already active before the outbreak of the COVID-19 pandemic and the remaining three were specifically created for sharing dreams during the first lockdown imposed by the World Health Organization between 17 March and 22 May 2020. From the total dream cohort presented in these platforms, we filtered out dreams that were shorter than a few lines, leaving us with a corpus of 120 dreams suitable for analysis from which a sample of 30 dreams was randomly selected. 

### 2.2. The Research Group Structure

The research group included six researchers: two supervising clinical psychologists, one of whom is also a group analysis scholar, two clinical psychology interns, one graduate student in clinical psychology, and, in order to gain a sociocultural perspective and a qualitative perspective, one anthropologist. This structure was designed to enable an integration between theoretical knowledge, clinical knowledge, and empirical experience. Nineteen group meetings were held, during which one of the researchers wrote a protocol detailing the main themes of group interactions and processes.

### 2.3. Data Analysis

The data were analyzed by applying a qualitative dream content analysis process. Transcripts were read as hard copies by the research group members. Initial readings were made aloud in order to hear the dreams in a more vivid way and to inhabit the reflective posture of in-dwelling more strongly [38]. Notes were made of the core concepts of each dream and the research group members’ reactions and emotions. During subsequent rereads two layers emerged: groups dream content analysis and the analysis of group processes and experiences. 

#### 2.3.1. Group Dream Content Analysis 

Following Kron et al. [9], the research group analyzed the dreams looking for recurrent central themes which can be related to the continuous stressful situation. In the first stage, we extracted codes from the data and used them as the basis for creating categories and sub-categories (Table A1, Appendix A). In the second stage, we calculated the percentage of occurrences of each category out of the total number of dreams. From an initial long list of categories, those finally selected appeared in at least 40% of the total sample of 30 dreams. 

#### 2.3.2. Analysis of Group Processes and Experiences 

The research group conducted and documented an open reflexive discourse regarding the processes and experiences that arose during the analysis. This open process enabled us to extract important themes and insights that the structured content analysis did not. Following Creswell et al. [39], we then merged the categories and the sub-categories and formulated three main themes. All members of the group research were involved in debating, co-constructing, and finalizing the interpretation of the dreams.

## 3. Results

The dream content analysis yielded three meaningful and coherent themes: (1) dominant threats: enemy, danger, and COVID-19; (2) emotional fusion: confusion and despair alongside recovery and hope; and (3) group processes characterized by movement between being alone and being together. 

### 3.1. Dominant Threats: Enemy, Danger, and COVID-19 

As expected, many of the SNS lockdown dreams referred to coping with hazards related to this period, referring either implicitly to an enemy, danger, or risk or explicitly to the COVID-19 pandemic. Many of the dreams related vicariously to the situation, and thus the dominant threat was portrayed as an enemy, namely ordinary humans from whom the dreamer needed to escape. For example, one dream described how the enemy appears and ends up killing the dreamer:

*I was jumping up trying to bat it [a key] out from underneath when a woman walked up**……She said, “I’ll get that for you” and knocked it down with a stick, except what she was holding wasn’t a key but some sort of syringe. Her expression changed and she suddenly slammed this thing into my forehead with a wry, cruel smirk and injected something into me: “Didn’t expect that did you, bitch?”. As I fell back, I knew I was dying*.

Some dreams portrayed the unexpected health or medical risks threatening the dreamer: “*And in the middle of setting it up…my co-worker had a severe medical issue, possibly a heart attack*”. Others associated the danger and threat with elements of the natural environment (e.g., water, fire, earth, or animals). In such dreams, the dreamer described a struggle with forces greater and stronger than themself and the danger of falling: “*The tiger was coming through the window”; “I was driving down a winding road in heavy rain. The windshield wipers were mesmerizing”; “The road had no shoulder; there were only steep cliffs down to the sea”*.

In the explicit COVID-19 dreams, threats featured as the risk of contagion when the dreamer was caught up in a crowd gathering, often with people who were not adhering to the social distancing regulations:

*I was taking a walk and groups of people kept coming closer and closer**……My attempts to ask them to move away felt like wind rather than words*.

*Next, I was in a room of people talking to a man who started coughing, and during our conversation we figured out that he had contracted the virus and had now possibly infected everyone in the room…I felt scared…not knowing what to do next*.

All these quotes indeed demonstrate the emergence in dreams of reactions evoked by the ongoing health threat.

### 3.2. Emotional Fusion: Confusion and Despair Alongside Recovery and Hope 

Dreamers manifested an emotional fusion which comprised mostly negative emotions but, nonetheless, also quite a few positive feelings. This demonstrates the relational fusion and confusion which has been found dominant during traumatic times [40]. The explicit negative feelings related to dreamers’ concrete lived experiences during lockdown: “*I began to panic…for some reason this discovery [of the chaotic reality] filled me with total horror”; “I was concerned about the locking up*”.

Other negative emotions of despair, helplessness, and confusion were also prevalent: *“I watch [the goose] desperately hoping it can fly up, but it’s clear it can’t. I’m very sad”; “She was desperately looking for someone to sub that class…. The conversation between us ended, and I went back to the yoga room feeling upset”; “The landscape is confusing to us”*. In some dreams, these emotions were accompanied by descriptions of damage and destructiveness: *“I looked around the ruined, out-of-control room …somehow he ripped all the wall around it …which he had opened up in a completely destructive way”*.

Positive feelings represented either success despite the difficulties and threat—*“I make it over the rough earth”*—or the ability to gain pleasure from things that the dreamer still feels in control of—*“I’m having fun getting in the holiday spirit though”*. Likewise, family members were also sometimes depicted positively. Examples included references to marriage and connections with parents: *“I was sitting at my own wedding…all I could remember was my mum smiling at me when I asked her who I was marrying*”. The following quote may be seen to demonstrate both the wish for regeneration and the ability to recover: *“The bird has recovered…I’m happy that it has recovered enough to return to the wilderness”*. Finally, an explicit sense of victory was also expressed: *“I awoke feeling like I’d truly defeated something and feeling really happy and regenerated”; “I woke up very happy”*.

### 3.3. Group Processes Characterized by Movement between Being Alone and Being Together

Many dreams expressed feelings of being alone and isolated from others: *“I am walking around a small city by myself at night…. There were no other people*”. Some exemplified the feeling of being distanced from others in more indirect metaphoric ways: *“I start to drive around and find that large piles of rough earth have been placed at and around the exit, preventing easy access to the main road”*. Beyond the fact that the dreamer is stuck and alone, the metaphor of the “main road” can be interpreted as a place with the presence of other people in contrast to the lonely place where the dreamer is located.

A desire for the presence of others but the impossibility of these relationships was manifested in many dreams: *“I saw something move by the window…it was a little girl but I couldn’t see her face so I moved a bit closer and it moved back…so I couldn’t see the face”*. Longing for others but being distanced from them while also sensing the danger accompanying these relationships was also illustrated in the following segment of the same dream: *“I am standing looking at a Noah’s Ark-type huge wooden boat, it’s empty in front of me. I woke up so sad that even in the safe space that dreams afford, I couldn’t experience closeness”*. The allusion to the Noah’s Ark myth represents both a catastrophe as well as repair and renewal. The ark, populated by couples, may be understood as a symbol of relationship vitality and rebirth [41]; yet, as it is empty, it exemplifies the aspect of danger coming from interpersonal contact as well as the fear that, on leaving the home that protected us during the flood, namely COVID-19, hazards would reoccur [42].

Other dreams related more explicitly to the desire for interpersonal contact as a protection against anxiety and as a source of hope: *“Meeting with friends on a grassy hilltop each of us wearing a gold ring with a mechanical bumblebee attached. We held our hands together, and the bumblebees went buzzing off with hope”*. This movement between being alone and being together describes the dreamers’ growing ability to experience ambivalence toward others (both in reality and in internal representations), which is considered the beginning of adequate mental health [43]. The ability to experience such emotionality enables the dreamer to find a possible coping mechanism: 

*Then I found myself driving through the streets…On the way out of town I saw a person in need of help, so I stopped and got out. More people came asking for my help, so I gave them what I could find in the car (blanket, mat, water bottle, etc.) and said I wished I had more to give, but they were grateful for my offerings. At some point, I also remember seeing children swimming with dolphins and a rainbow in the distance*.

While coping with a threatening episode, the dreamer derives comfort, protection, and potency from belonging to others and saving those who are in need. This results in a more hopeful, albeit fantastical, ending. 

Some dreams exemplified recurrent manifestations of omnipotence and magical experiences: 

*I was woken up by God and told I’m in the part after the end of the world; I realize it would be faster to fly, so I put my arms by my sides with my fingers out and palms pointing to the earth which helps me levitate. It’s been a while since I’ve flown so it takes me time to remember how to lift myself high over the tree line*.

These manifestations may be understood as coping mechanisms; as they are beyond everyday threats and reality, they provide identity, hope, and relief. 

In addition to the themes extracted from the content of the dreams, we also formulated two more sub-themes through the research group reflexive analysis regarding the processes and experiences of movement between being alone and being together: oscillation between fragmented and coherent structures and oscillation between rejection and belonging.

#### 3.3.1. Oscillation between Fragmented and Coherent Structures

During the research group members’ encounters with each dream, a chaotic “Tower of Babel” feeling arose, as if the dreams were attacking our ability to think and understand. Reactions to the initial dream readings were characterized by bewilderment—“what is this nonsense?”—as well as a sense of fragmentation and despair—“nothing will come out of it”. This tension between fragmented and coherent structures was also expressed in our discussions regarding the analyzing method of the dreams, which ranged from structured dream content analysis to a free and interpretive associative discussion. Some group members expressed discomfort with the structured method, fearing it might reduce richness and complexity; others expressed difficulty with the “chaos” and lack of boundaries involved in the open discussion which was perceived as preventing “scientific” analysis. 

We found ourselves immediately alluding to the myth of Babel, thus illustrating Bion’s notion of the attack on linking, according to which the link between the individual and the group may come under attack from different directions [35]. However, after both the content and reflexive discussions of the dreams, the opposite feeling emerged, namely a sense of the creation that appears out of the chaos, accompanied by a sense of gratification.

#### 3.3.2. Oscillation between Rejection and Belonging

During research group meetings there was rapid movement between experiences of agreement and belonging, on the one hand, and a sense of total disagreement, on the other hand. Although the research group’s crystallization process led to an increased sense of togetherness and belonging to a small community, fears of exclusion and rejection were also apparent. The inevitable disagreements between members were experienced at times as risky, leading to over-involvement with its accompanying fear of excessive exposure and exclusion. As central features of resonance, these emotions were understood to represent strong identification with, to return to the earlier allusion, both those who were invited into Noah’s Ark and those whom Noah had to reject [44].

## 4. Discussion

The aim of the current study was to explore oneiric life shared on SNS during the first COVID-19 lockdown. We used an epistemological group-analytic approach to conduct content analysis and investigate unique group processes. The findings demonstrate dominant expressions of threats and dangers, such as enemies and COVID-19. Furthermore, the emotional fusion of both negative emotions, such as confusion and despair, and positive feelings such as recovery and hope was detected. The unique group processes produced by both the content and the research group’s reflexive analyses were characterized by the movement between being alone and being together, between fragmented and coherent structures, and between rejection and belonging. 

Generally, the resulting first two themes resemble current pandemic dream research which testified to the presence of expressions of threatening events and death in general [45] and, specifically, of references to COVID-19 [12,13,46,47], such as a lack of social distancing or actual infection. Of note, the threat of the natural environment, humans, or the virus, for example, was portrayed as an enemy, thus using the war-like setting metaphor common to descriptions of illness in general [13,48]. The pandemic was portrayed as a lethal enemy and health experts were portrayed as *“heroes fighting on the front line in a battle*” [12,49]. Moreover, the challenge relating to the struggle with forces greater and stronger than the dreamer (e.g., wild nature, animals, falling) may have represented efforts to describe more profoundly the threats and terror individuals had to cope with during this period [50]. 

As expected, the explicit reference to COVID-19 featured the dreamer or a person close to them at risk of contagion from an unfamiliar stranger or a crowd acting malevolently or thoughtlessly and stressed the need to escape [13,47,51]. Yet, in line with findings from a recent systemic review of the literature on dreaming during the COVID-19 pandemic [52], the percentage of dreams with explicit pandemic content was smaller than expected, common in only 12% of dream narratives. 

Our findings regarding the emotional content of dreams correspond with recent research on dreams during COVID-19, which found negative emotions, such as sadness, depression, confusion, anxiety, and the experience of helplessness, to be more prevalent than positive emotions, e.g., [13,51,53]. Psychoanalytic theory, e.g., [54] and trauma research, e.g., [55,56] have defined helplessness, avoidance, and confusion symptoms [57] as illustrating the shattering emotional consequences of trauma. Hence these emotions were highly present in the dream experiences during the COVID-19 lockdowns. This congruence between the emotional experiences manifested in the dream content and the characteristics of the external reality corresponds to the continuity hypothesis [58] and the threat simulation theory [36]. Accordingly, in times of crisis such as the COVID-19 outbreak, dreaming about external threats supports the ability to deal with the crisis and helps to relieve mental suffering, e.g., [52,59]. 

However, in the present study, alongside the dominant negative feelings, the dreams also included positive and pleasant emotions of success, fun, and recovery. This striving for competence and regeneration may reflect the dreamers’ conscious and subconscious want and wish to be freed from isolation and to return to known forms of interpersonal and physical closeness [53]. It may also reveal their coping attempts and level of inclusion [60] and their hopes for the future [61]. 

### 4.1. Collective Trauma—The Group-Analytic Perspective

From a group-analytic perspective, our findings further show dreams to reflect a collective cultural product [1,27,49] which can thus serve as a tool for identifying the unique social and psychological group processes manifested during times of collective trauma and natural disasters [22,40]. Following, specifically, trauma in the social sphere, the fear of annihilation caused by inadequate holding and associated with the failed dependency of the leaders [40] may be characterized by oscillation between aggregation (i.e., a person relies on themself and gives up on society as a source of support) and massification (i.e., the group denies the differences between people and therefore belonging to the group provides a sense of omnipotence) [22,40]. 

Experiences of loss, abandonment, and damage are, therefore, associated with anxieties of fragmentation in oscillation with anxieties regarding relational fusion and confusion [22]. For example, the use in both the dynamic matrix of SNS and the research group of metaphorical and symbolic images such as the Babel myth—the earliest biblical account of an anonymous, task-oriented, egalitarian, and leaderless group activity [62]—represents both persecution and fragmentation anxieties as well as phantasies of omnipotence [63]. These omnipotent wishes accompanied by magical measures, such as flying, may be understood as a mechanism used by traumatized people in order to cope with annihilation anxieties evoked by social trauma. This mechanism is, in turn, used as a magical wish for conquering the fear and a means to restore, when felt to be threatened, a lost sense of worth and social identity [22,40]. The prevalence of positive emotions as well as the omnipotent wishes may also be conceptualized as an aspiration to glory, which is not only a defensive phenomenon but also a primary impulse [64]. By idealizing and emulating heroes and events whose mental representations elicit a shared feeling of success and triumph (e.g., Noah’s Ark), one fulfills a sense of self-value and belonging [65]. Glory can thus be perceived similarly to renewal, hope, and faith—all helpful components of the therapeutic work with trauma [60]. 

### 4.2. The Transformative Potential of Dreamtelling on SNS

The emergence of the COVID-19 pandemic and the need for social distancing, which led to face-to-face group meetings being replaced with virtual settings worldwide, motivated many people to share dreams in designated groups. This suggests both their unconscious requests for containment from others and their efforts or demands for influence on them [24]. Specifically, the dynamic matrix of SNS served as a vast container which matched the containment needs of individual dreamers by serving as an unconscious effort to continue to work through the others [24,26]. The SNS dynamic matrix enabled the individual dreamer to share a dream with the group in order to further the containment and elaboration of their dominant personal and social anxieties. This dream sharing served as a kind of “transitional space” [66]: a dream that comes from nowhere and from no one that invites participants to share their differences and agreements in a common dynamic space which is crucial for individual and social growth [35]. The attempt to meet others through emotions of loss thus allowed both dreamers and their audience to adapt to change and, possibly, to start mourning the painful feelings associated with massive collective trauma [47]. This working-through enhanced the individual capacity for symbolization and fostered hope that these anxieties could be both tolerated and further processed [1,26]. 

The unconscious request to influence others (the dream audience) was expressed by the dreamers’ need to be released from the isolation and inactivity forced by the lockdown and to be part of or be with others who share similar but also different experiences. The exchange, mirroring, and resonance reciprocal responses strengthened the dreamers’ and the audience’s sense of belonging and enabled a benign transformation from loneliness and personal isolation into a more meaningful dialogue [34]. Moreover, as SNS are defined as having no center or sidelines, no hierarchy, and no central leadership, they provide an open horizontal structure as an alternative to the disappointing closed hierarchical structures of the existing governing and leadership systems [67]. SNS thus exemplify how the free communication of dreamtelling may enable general feelings of personal and social glory, even in the face of trauma [60].

The use of group-analytic phenomenology allowed us-authors and research group members- to gain insight into the emotional experience of the group and to better understand how our individual emotions were connected to the collective experience of the group. This helped us to find a sense of belonging and support within the group, which provided a sense of containment and helped us to navigate the challenging circumstances of the pandemic. Moreover, this phenomenology enabled us to translate [23] what had begun with expressions of emotional confusion and chaos and then split into a beneficial reflection of rich emotional experience that produced two main results. First, the contact with the content and emotions of the dreams shared on SNS enlarged our perception of collective human responses to major traumas [68]. Second, the creative social connections forged within the group enabled the digestion and relief of mental pain. Indeed, the unique social and psychological safe space for sharing and discussing the dreams served as a kind of ark that enabled all the researchers to both comply with the dreamers’ unconscious request for containment and to reconnect to the wider society via the dream wisdom—the wisdom of the unconscious—that is necessary for the survival of the human race [69]. 

### 4.3. Limitations

Aside from these contributions, the present study has several limitations that should be taken into consideration. First, the cross-sectional research design did not enable an examination of thematic changes over time (i.e., before, during, and after the pandemic). Second, we specifically chose dreams that were rich enough to allow us to observe the functions of the dream and the attitudes toward the dreams. Third, factors associated with different motives for sharing dreams were not observed. This selection process and the fact that we used dreams shared online might have biased the results toward dreams with richer content shared by people with a particular interest in dream sharing and psychology. 

Nonetheless, the results of the present study suggest that further investigation, using both qualitative and quantitative methods, is needed to explore the dynamics of group dreamtelling in various contexts, including different settings, conflicts, cultures, and groups of people. Moreover, longitudinal dreamtelling studies including both psychophysiological measures such as heart rate variability or the tonic/phasic skin conductance, and psychological measures evaluating the motives for sharing dreams [70] may further broaden this approach and minimize retrospective bias.

## 5. Conclusions

The findings of the current study demonstrate the nature of dreams as both an individual and a collective cultural product that enables the exploration and detection of the unique social and psychological group processes manifested during times of stress and collective trauma. Furthermore, studying dreams shared on SNS showed how this matrix enabled the individual dreamer to further the containment of dominant personal and social anxieties while discovering new multipath routes back to the wider society. This illuminates once more the transformative potential of dreamtelling and the importance of group psychology frameworks for studying human reactions to major traumas and their role in contemporary societal movements. We hope that these findings will stimulate further group-oriented dream research on, in particular, collective traumatic events and the function of dreams to facilitate affective coping. 

## Data Availability

The datasets used and analyzed in the current study are available upon request from the corresponding author.

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
