# Peer review of "Dreams Shared on Social Networks during the COVID-19 Pandemic: A Tower of Babel or Noah’s Ark?—A Group-Analytic Perspective"

_ijerph, 2023, doi:10.3390/ijerph20043534_

Round 1

Reviewer 1 Report

The paper is interesting and well written. I do not have any specific suggestion to improve it.

Nonetheless , being I a physiologist, I am not familiar with such kind of studies and papers. I might wonder , in fact,  whether the reported data could be supported by psychophysiological meausures . In this case, the authors may wish to indicate which variables could be studied, at least in  further research, to show, for instance, that the subjective, positive or negative elaboration of shared dreams increased or decreased the heart rate variability or the tonic/phasic skin conductance or reaction times to neutral/pleasant/unpleasant images and so on.

Author Response

Reviewer # 1:

The paper is interesting and well written. I do not have any specific suggestion to improve it.

We thank the reviewer for the effort invested in reviewing our paper and for their feedback and support.

  1. Nonetheless, being a physiologist, I am not familiar with such kind of studies and papers. I might wonder, in fact, whether the reported data could be supported by psychophysiological measures. In this case, the authors may wish to indicate which variables could be studied, at least in further research, to show, for instance, that the subjective, positive, or negative elaboration of shared dreams increased or decreased the heart rate variability or the tonic/phasic skin conductance or reaction times to neutral/pleasant/unpleasant images and so on.

As it is a qualitative study of dreams shared on SNS, it is focused on the fact that while no one is able to escape the impact of the ongoing pandemic, sharing it in a group can ease the burden and enhance coping possibilities. This was measured, as common in such studies, by the analysis of the subjective experiences of the research group members. Nevertheless, it may be interesting to conduct future research which will include psychophysiological measures such as heart rate variability or the tonic/phasic skin conductance, before and after the dreamtelling process. This idea has now been added to the future studies section (please see page 9). 

Reviewer 2 Report

The topic of the paper is very interesting, but there should be made some alterations to increase the quality of the paper.

1. Eventhough the authors used a qualitative approach to analyze COVID-19-related dreams, they should give a comprehensive overview what is known about dreams during the COVID-19 pandemic, that is, review the quantitative empirical research.

2. The theories about possible dream functions are not well presented, there is a lot more to it compared to the references the authors included, e.g., citing the TST in the context of stressful times. The current theories should be presented in a comprehensive way, e.g. Social Simulation Theory, Constructive episodic simulation hypothesis and so on.

3. I have problems with understanding the relevance of the four phases presented in "1.1. The Group-Analytic Approach to Dream Sharing". It seems obvious that one have to dream about it, remember the dreams, and record it in order to share it with others. What does "when the digestion is not satisfactory" mean? Here is the major problem with the study, the dreams are self-selected, what dreams are published on social network sites, what are the persons motive to do so. This should be discussed in detail in the discussion. The authors can consult: Graf, D., Schredl, M., & Göritz, A. S. (2021). Frequency and Motives of Sharing Dreams: Personality Correlates. Personality and Individual Differences, 175, 110699. https://doi.org/10.1016/j.paid.2021.110699

4. I have problems in the sentence "the surprise of COVID-19 – a unique traumatic event and a threat on both the individual and the society" with defining the pandemic as a traumatic event, research in PTSD clearly uses another definition regarding traumatic event.

Author Response

Reviewer # 2:

The topic of the paper is very interesting, but there should be made some alterations to increase the quality of the paper.

We thank the reviewer for reading our manuscript and offering suggestions for its improvement.

  1. Even though the authors used a qualitative approach to analyze COVID-19-related dreams, they should give a comprehensive overview what is known about dreams during the COVID-19 pandemic, that is, review the quantitative empirical research.

Thank you for this suggestion. We have now included additional overview for what is known about dreams during the COVID-19 pandemic, referring to recent systematic reviews (please see pages 1-2).

  1. The theories about possible dream functions are not well presented, there is a lot more to it compared to the references the authors included, e.g., citing the TST in the context of stressful times. The current theories should be presented in a comprehensive way, e.g. Social Simulation Theory, Constructive episodic simulation hypothesis and so on.

With respect to this reviewer, the aim of the paper was to examine dreams shared on social network sites (SNS) during the first COVID-19 lockdown, applying a group-analytic approach, which is the only theory that refers to dreams in a group or social context. All other theories refer to the individual experience.  Therefore, we included such theories in the relevant section of the discussion (please see p. 7).

  1. I have problems with understanding the relevance of the four phases presented in "1.1. The Group-Analytic Approach to Dream Sharing". It seems obvious that one have to dream about it, remember the dreams, and record it in order to share it with others. What does "when the digestion is not satisfactory" mean? Here is the major problem with the study, the dreams are self-selected, what dreams are published on social network sites, what are the persons motive to do so. This should be discussed in detail in the discussion. The authors can consult: Graf, D., Schredl, M., & Göritz, A. S. (2021). Frequency and Motives of Sharing Dreams: Personality Correlates. Personality and Individual Differences, 175, 110699. https://doi.org/10.1016/j.paid.2021.110699.

Thank you for raising these subjects. We've added an explanation regarding the digestion in the fourth phase to the introduction section, and hope that that it is now better described (please see page 3). Furthermore, we have added and referenced the motivation of persons to share their dreams on SNS in the limitations section on page 9 as requested.

  1. I have problems in the sentence "the surprise of COVID-19 – a unique traumatic event and a threat on both the individual and the society" with defining the pandemic as a traumatic event, research in PTSD clearly uses another definition regarding traumatic event.

Please see our response above, our intention was not to study PTSD in specific, but to describe people’s main experiences in times of collective trauma by analyzing their dreams shared on social network sites (SNS) applying a group-analytic approach. The description of COVID-19 as a traumatic event and a threat on both the individual and the society follows in recent studies like (the first two papers are now included in the manuscript):

Brooks, S. K.; Webster, R. K.; Smith, L. E.; Woodland, L.; Wessely, S.; Greenberg, N.; Rubin, G. J. The psychological impact of quarantine and how to reduce it: Rapid review of the evidence. Lancet 2020, 395(10227), 912–920  

Mariani, R., Gennaro, A., Monaco, S., Di Trani, M., & Salvatore, S. (2021). Narratives of dreams and waking thoughts: Emotional processing in relation to the COVID-19 pandemic. Frontiers in Psychology12, 745081.

Marogna, C., Montanari, E., Contiero, S., & Lleshi, K. (2021). Dreaming during COVID-19: the effects of a world trauma. Research in Psychotherapy: Psychopathology, Process, and Outcome24(2).